# Association of Individual and Contextual Factors with Chronic Spine Problems: An Analysis from the National Health Survey

**DOI:** 10.3390/ijerph22060879

**Published:** 2025-05-31

**Authors:** Aryostennes Miquéias da Silva Ferreira, Sanderson José Costa de Assis, Clécio Gabriel de Souza, Geronimo José Bouzas Sanchis, Rebeca Freitas de Oliveira Nunes, Marcello Barbosa Otoni Gonçalves Guedes, Johnnatas Mikael Lopes, Angelo Giuseppe Roncalli

**Affiliations:** 1Faculty of Health Sciences of Trairi, Federal University of Rio Grande do Norte, Santa Cruz 59200-000, RN, Brazil; aryostennes@gmail.com (A.M.d.S.F.); clecio.gabriel@ufrn.br (C.G.d.S.); rebecafdon@gmail.com (R.F.d.O.N.); 2Department of Dentistry, Federal University of Rio Grande do Norte, Natal 59078-900, RN, Brazil; gero.bouzas@gmail.com (G.J.B.S.); angelo.oliveira@ufrn.br (A.G.R.); 3Department of Physiotherapy, Federal University of Rio Grande do Norte, Natal 59078-900, RN, Brazil; marcello.guedes@ufrn.br; 4Department of Medicine, Federal University of Vale do São Francisco, Paulo Afonso 56304-917, BA, Brazil; johnnatas.lopes@univasf.edu.br

**Keywords:** spinal diseases, social determinants of health, multilevel analysis, health surveys cross-sectional studies

## Abstract

The spine is the most affected region, which compromises functionality and generates absenteeism, increased health care costs, and disability retirement rates. Based on the biopsychosocial model, it is believed that chronic back problems are the result of a complex network of factors, both individual and contextual. A cross-sectional study was developed with data from the 2013 National Health Survey, the United Nations Development Programme, and the National Register of Health Establishments (state level) for the second and third levels of aggregation, respectively. Multilevel Poisson regression was performed at three levels. The prevalence of chronic back problems was 18.5% (95% CI 17.8; 19.1), with a higher prevalence in females (RP = 1.23; 95% CI 1.15; 1.30), those aged above 49 years (RP = 1.75; 95% CI 1.61; 1.90), those performing heavy activities at work (RP = 1.37; 95% CI 1.28; 1.46), those with depressive days (RP = 1.70; 95% CI 1.50; 1.94), those who were smokers (RP = 1.37; 95% CI 1.27; 1.48), and those in states with a higher coefficient of Family Health Support Team per 100,000 inhabitants (PR = 1.28; 95% CI 1.07; 1.54). Chronic spine problems were associated with biological and behavioral factors and were more strongly associated with the coefficient of Family Health Support Team in Brazilian municipalities.

## 1. Background

Persistent pain for at least 6 months is a frequent health problem that affects all age groups; it is responsible for 7.7% of years lived with disability (YLDs) and affects different regions of the body, such as the lower and upper limbs and spine [1,2]. The spine is the most affected region and can cause disability, absenteeism, and increased rates of disability retirement, in addition to increasing healthcare costs such as those associated with surgery, physical therapy, and complex procedures [3,4].

The prevalence of chronic back pain in Brazilian adults in 2013 was approximately 20%. It is estimated that 70 to 85% of the population will have a back pain episode at least once in their lifetime [3]. Chronic spine problems have been described through a biopsychosocial model in which the risk factors for this condition are determined by the interaction of individual, psychological, and social issues [5,6,7]. Based on this framework, chronic spine problems are believed to be the result of a complex network of aspects, both individual and contextual.

The development and promotion of health are associated with a relationship between the social environment and how people relate to their social context [8]. In chronic diseases, this context is no different, and in Brazil, chronic spine problems have been associated with socioeconomic factors, such as low education levels, living in urban areas [3], and social inequality [9,10]. Thus, the relationships among these factors can be observed through social health determinants, in which the living and working conditions of population groups are related to their health status [10]. Health determinants comprehend the combined effects of physical and social factors on individuals and communities. People and populations’ health are influenced by a wide range of factors, which can be classified as contextual (those related to environmental and socio-economics, such as living conditions, employment rates, average income, and access to health services); or individual (referring to a healthcare system user’s personal aspects, such as genetic factors, educational level, lifestyle, attitudes, and beliefs). Analyzing these factors makes it possible to understand the health tendencies of a population group and the differences in health conditions between diverse social groups that constitute a specific community [10].

In 2013, the National Health Survey (NHS) was conducted, in which chronic spine problems, among other factors, were investigated. However, few studies have analyzed the associations of chronic spine problems with individual and contextual factors, with two levels of aggregation across linked public databases. Chronic back pain is a multifactorial condition, the manifestation and intensity of which are modulated by the complex interaction between individual and contextual factors. Among the individual factors, biological characteristics such as age, sex, genetic predisposition, and the presence of comorbidities stand out, in addition to psychosocial aspects such as beliefs, emotions, and coping strategies. At the same time, contextual factors such as the work environment, socioeconomic level, and access to health services and social support play a determining role in the modulation of pain and the functional capacity of the individual. Thus, an understanding of chronic back pain must transcend the exclusively biomedical perspective, incorporating a biopsychosocial approach that recognizes how these multiple dimensions interact and influence pain and its impact on daily life.

Since it is considered a highly prevalent problem and a major disability issue worldwide, as well as because of the influence of contextual and individual factors and healthy living environments with access to healthcare services to support daily activities, in addition, to date, no studies have been identified that analyze the interaction between individual and contextual factors that may influence chronic spinal problems. This study aims to investigate how both individual-level (such as age, sex, and occupation) and contextual-level (such as state-level health service coverage) factors are associated with the prevalence of chronic spine problems among Brazilian adults, using data from the 2013 National Health Survey.

## 2. Materials and Methods

### 2.1. Study Design

This was a cross-sectional study conducted from the Brazilian NHS (2013) database. The main outcome was “spinal problems”, and the independent variables were the individual and contextual factors. This database was linked to another database with a state level of aggregation, such as the 2010 National Census, conducted by the Brazilian Institute of Geography and Statistics (IBGE), whose data were compiled by the Brazilian Agency of the United Nations Development Programme (UNDP), in which some indicators were created, such as the Human Development Index (HDI) and the Gini coefficient, among others, and the National Register of Health Establishments (CNES), with information on public and private health services for the entire country.

The population to be surveyed corresponds to residents of private households in Brazil, except those located in special census sectors (barracks, military bases, lodgings, camps, vessels, penitentiaries, penal colonies, prisons, jails, asylums, orphanages, convents, and hospitals).

### 2.2. Context and Research Participants

At the first level of aggregation, only two variables were included: sex and age group. At the second level, behavioral variables were considered: heavy work activity, perceived health status, number of days feeling depressed, and smoking addiction.

A national health survey in Brazil aimed to characterize the health status and lifestyle of the Brazilian population, as well as aspects related to healthcare [10]. Complex conglomerate sampling was carried out in three stages: the first stage involved census sectors, the second stage involved households, and the third stage involved residents aged 18 years or older. For this reason, the survey has sample weights for each census sector, for the households, and for the interviewees. In the end, 64,348 households were visited, and 60,202 interviews were carried out, constituting the final sample of the study. All analyses were conducted considering the complex sampling design of the 2013 PNS, using the Complex Sample module of SPSS software (V22.0) and the Survey Data Analysis command (svy prefix) in Stata 13.0, in order to correct for the effect that the clustering of primary sampling units has on the estimates, known as the design effect (DEFF).

### 2.3. Variables and Data Sources

The outcome variable of the present study was measured by the National Health Survey Questionnaire using the following question: “Do you have chronic spinal problems, such as chronic back or neck pain, low back pain, sciatica, and vertebral or disc problems?”, with response options of ‘yes’ or ‘no’.

The following independent variables were analyzed:1.Individual demographics
a.Sex (male and female)b.Age (up to 33 years, 34 to 48 years, and 49 years or older)2.Individuals—Lifestyle
a.Heavy activity at work (yes or no)b.Health status evaluation (very good, good, fair, bad, or very bad)c.Depressive days (none of the days, fewer than half of the days, more than half of the days, and almost every day)d.Smoking (never smoked, less than daily, and daily)3.Contextual—Socioeconomic
a.Gini coefficient (above 0.735, from 0.675 to 0.734, and up to 0.674)b.Human Development Index (HDI) (above 0.735, from 0.675 to 0.734, and up to 0.674)c.Sociodemographic conditions (low, intermediate, and high)4.Contextual—Health Services
a.Coefficient of Family Health Support Team per 100 thousand inhabitants (≤0.946; 0.947 to 1.876; ≥1.876)

### 2.4. Quantitative Variables

The quantitative variables were categorized into three strata based on the tertiles of each of the variables. The study’s independent variables were selected based on a theoretical model developed by the authors based on the current literature and the plausibility of the association with chronic spine problems.

### 2.5. Statistical Methods

The social determinants of the health model used in this study were proposed by Dalgren and Whitehead [11] and are based on layers from specific levels of determination. The first level: individual-level determinants, such as age, race, sex, and genetic factors. The second level: lifestyle factors, such as dietary and lifestyle habits, and another layer represented by social and community networks. The third level: general socioeconomic, cultural, and environmental conditions [11]. For the linkage of the databases, a deterministic linkage was performed based on the code of the state.

Poisson multilevel modeling was used in the statistical analysis to assess the influence of individual and contextual determinants on the dependent variable. In this analysis, the contextual level can be considered as social aggregates because of the effect on their members. Thus, subjects are generally considered the first level (lowest level), and the places where they live are considered the second level (outermost level) [8,11,12]. For analysis of the places where they lived, state-level data were taken into account, as they represented the smallest territorial units available for data analysis for this type of analysis by IBGE.

Data on sociodemographic factors were derived through exploratory factor analysis, using the following variables: total fertility rate, infant mortality, percentage of people vulnerable to poverty, per capita income, and formal employment status of individuals over 17 years old. These variables were selected based on their theoretical relevance to adequately represent sociodemographic conditions, as well as their statistical association with the outcome. Following the scree test, the Kaiser–Guttman criterion, and Horn’s parallel analysis, only one factor was extracted. This factor was subsequently categorized into three groups based on tertiles to facilitate interpretation and analysis [13].

Descriptive analysis was then performed to verify the cutoff points or criteria for categorization. Rao Scott’s and chi-square tests were performed between the outcome variable and all independent variables, and those with *p* values of 0.2 or less were selected for inclusion in the multiple regression. Unadjusted prevalence ratios (PRs) and their respective 95% confidence intervals (CIs) were first estimated. Next, a multilevel Poisson regression model was performed, including the variables from the different levels. Modeling was started with a null model to verify the feasibility of multilevel modeling, and then the variables of all dimensions were included, starting with the proximal variables, until the inclusion of the more distal variables. The model fit was performed using the likelihood ratio test, as well as by observing the change in variances between each model. The PRs were estimated by the proportions of exposed and unexposed individuals and their respective 95% confidence intervals. First, the individual-level variables were included, followed by the context variables and the health policy variables, which were included one by one at each level analyzed. The data analysis was conducted in the statistical program Stata 16.0. A significance level of 5% was adopted (α < 0.05). Although PNS is a complex sampling survey, with the inclusion of grouping variables and sample weights, in this study, for descriptive analysis, the sample weights were considered; however, they could not be considered in the association analysis because the probabilities required for each level were not available in the database made available by the IBGE.

When including variables in multilevel modeling, the null model was initially carried out, in which variables with characteristics at the individual level were included. Then, in Model 2, variables at the individual level were combined with those at the contextual level. Finally, in the final model, only the variables that fit the theoretical model and that were significant were maintained.

For the treatment of lost data, we followed the recommendations proposed by Hair Jr. et al. [14], in which the randomness of losses must be verified. Initially, the effect of lost data on the estimates was verified by checking the significance of the difference in some variables of interest when considering the presence or absence of data. Next, Little’s test for completely randomly lost data was applied, which showed a value >0.05, indicating that the losses occurred randomly.

Since this is a study with secondary public data, this research does not require a new submission to the analysis of ethical principles, considering that the PNS 2013 project was approved by the National Research Ethics Committee of the National Health Council [15].

## 3. Results

From the 2013 National Health Survey, 64,308 records of adults aged 18 or older were extracted, with a final analyzed sample of 60,202 individuals, representing the Brazilian adult population at the state level. Missing data were below 10% for all variables, with random distribution. The most common profile was women (56.9%), aged up to 33 years (33.7%), non-smokers (80%), not engaged in heavy work, and with a good self-perception of health (52.7%). Most reported no days of depression (77.6%), intermediate sociodemographic status (36.9%), and lived in areas with a Gini coefficient up to 0.56 (42.8%) and an average HDI up to 0.674 (45.3%). Additionally, most municipalities had up to 0.946 Family Health Support Teams per 100,000 inhabitants (Table 1).

However, among those who reported chronic spine problems, the highest prevalence was among women (19.3%), who were aged 49 or over (25.5%), who performed heavy work (21.2%) and who had a very poor health perception (44.3%). This population reported being depressed almost every day (36.1%), having a daily smoking habit (26%), and having a percentage of Family Health Support Team teams above 1.876 per 100,000 inhabitants (Table 1).

According to the adjusted multilevel model, we observed that women had worse chronic back problems than men did (PR = 1.23; 95% CI 1.15; 1.30), as did people of older ages (PR = 1.75; 95% CI 1.61; 1.90), who performed heavy activities at work, such as high physical demand (PR = 1.37; 95% CI 1.28; 1.46); who had worse self-assessments of health status (PR = 3.92; 95% CI 3.03; 5.07); and who had a greater frequency of depressive days (PR = 1.70; 95% CI 1.50; 1.94) and smoking (PR = 1.37; 95% CI 1.27; 1.48). In this analysis, no associations were detected between the context variables and the outcome variables, except for a greater percentage of Family Health Support Teams per 100,000 inhabitants (PR = 1.28; 95% CI 1.06–1.54) (Table 2).

## 4. Discussion

This study aims to investigate how both individual-level and contextual-level factors are associated with the prevalence of chronic spine problems among Brazilian adults, using data from the 2013 National Health Survey.

Regarding individual factors, such as sex, age, and level of physical activity, they were related to chronic back problems, but there was no combined association with socioeconomic factors. A higher prevalence of chronic back problems was observed in females, which corroborates the findings of other studies. Women historically have a higher prevalence of health problems; they combine their domestic activities with external work, which leads to greater physical stress and increases their susceptibility to health problems [3,5,16,17]. In addition, anatomical and functional issues, such as shorter height, lower muscle mass, lower bone mass, weaker joints, lower adaptation to physical effort, and greater fat weight, in addition to hormonal factors related to pregnancy, such as relaxin, estrogen, and progesterone, increase the propensity for biomechanical changes, predisposing individuals to back problems [3,16,17].

Regarding age, there is an association between advanced age and chronic back problems, which can be caused by different conditions, such as degeneration of the intervertebral disc, which is the most common cause of back problems, and postural problems, which can cause or increase pain [18,19].

Smoking has been shown to be increasingly associated with the outcome variable, with people who smoke more frequently per week presenting more chronic back problems. Nicotine, a substance present in tobacco, can activate the immune system, predisposing individuals to chronic conditions, such as chronic back problems [3,20,21].

Finally, a higher percentage of people with back problems was observed in those who performed heavy activities at work than in those who did not. Heavy work activities are linked to tissue overload and repetitive movements, which can predispose individuals to chronic injuries that are difficult to control [22]. Additionally, work-related musculoskeletal disorders are characterized by the overuse of the musculoskeletal system, which can result from repetitive exercise, continued use of the same muscle groups, and a lack of time for recovery, factors that are more common in individuals who perform heavy work activities [5].

Regarding contextual factors, such as social and economic characteristics, there were no significant associations to justify chronic back problems, despite their relevance to several other conditions and diseases. However, higher coefficients were observed for locations that had a Family Health Support Team program, showing a direct association between chronic back problems and the Family Health Support Team coefficient, as a contextual factor.

The inclusion of this program was implemented in Brazil through Ordinance No. 1154, of 24 January 2008, of the Ministry of Health. It is configured as a policy that works in collaboration with Family Health in Primary Health Care to increase its resolution [23]. The implementation of these multidisciplinary teams arises according to local health needs [24,25,26]; thus, the implementation of a Family Health Support Team as a public policy to improve primary care is directly related to the needs of the population and may be associated with chronic back problems, as demonstrated in the present study.

Therefore, it appears that where there is greater availability of health services with the presence of a Family Health Support Team, there is a higher prevalence of chronic back problems, not necessarily because chronic back problems occur more frequently but due to the greater demand for services due to their availability, which generates records of health conditions for the target population. However, an analysis is needed to evaluate this possible causal association, which cannot be answered through this study.

The Family Health Support Team coefficient was considered a factor associated with chronic back problems, probably because it is a pent-up demand and the teams become the place where these cases are detected. It is certainly not that the problem did not exist in places where the coefficient was lower, but it is probably a health demand of the population that was underreported. In the future, it is expected that, with the valorization of multidisciplinary teams in primary care, there will be changes in the epidemiological profile of the Brazilian population through the greater reliability of the data available to the health system, in addition to being a driver of the implementation of public health policies.

The Gini coefficient, a measure of social inequality, did not show a significant association with chronic back problems. Although Brazil has better indicators in the South and Southeast regions, income inequality is still uniform throughout the country, and this inequality seems to be more affected by national economic policies than by state policies [27]. In this reference, Wilkinson and Pickett [28] point out that the association between income inequality and health problems is more likely to be found in studies involving large areas, in which the unit of analysis is the country, for example, which may explain the lack of association in this study. In addition to the factors analyzed, no variability was observed in the health status outcome, and with these analyses, other responses were observed.

One of the limitations of this study is the way the outcome variable was assessed, since it was self-reported, which may contain information bias, although this can be minimized, since the error is likely to be randomly distributed among the analysis groups, which does not interfere with the interpretation of the results. In addition, the design of this study does not allow for inferences of causality, since it is a cross-sectional study; therefore, variables that may present reverse causality should be observed with caution.

Other limitations that can be highlighted in this study are the way in which the multilevel modeling was performed using the country as the unit of analysis due to the availability of indicator variables only at this level of aggregation. However, considering the size and diversity of the country, it is known that this unit of aggregation is not sustainable, since there is still considerable variation in context within it. The theoretical explanatory model assumed in this study can be considered limited, given the complexity of the factors that affect the social context of individuals and their perceptions, since there are unknown factors involved in determining the outcomes that could not be included in this approach.

It is also worth considering other aspects that could not be assessed in this study, such as the perception of professionals in the Family Health Support Teams and the reasons for their non-existence in some locations, which depend on management.

A follow-up study that controls for the individual, specific factors that should be monitored, such as levels of depression and workload, and contextual variables, presented in this study, would allow for a more accurate assessment of the factors that are related to the development of chronic back problems.

## 5. Conclusions

Thus, it was verified that in addition to individual biological and behavioral factors, chronic back problems can be determined by structural issues and the availability of services, such as a higher coefficient of the Family Health Support Team in the municipalities. However, in Brazil, these problems seem not to be influenced by factors that affect the social context.

## Figures and Tables

**Table 1 ijerph-22-00879-t001:** Descriptive analysis of variables and chronic spine problem distribution with the independent variables. (N = 60,202).

	Total (%)	Chronic Spine Disease
		n (%)
Sex		
Male	25.920 (43.1)	3.957 (15.3)
Female	34.282 (56.9)	6.621 (19.3)
Age (years)		
≤33	20.263 (33.7)	1.992 (9.8)
34 to 48	19.732 (32.8)	3.428 (17.4)
≥49	20.207 (33.5)	5.158 (25.5)
Heavy work activity		
No	27.182 (75.5)	3.739 (13.8)
Yes	9.260 (25.5)	1.959 (21.2)
Health state self perception		
Very good	7.433 (12.3)	596 (8.0)
Good	31.708 (52.7)	3.877 (12.2)
Regular	17.197 (28.6)	4.530 (26.3)
Bad	3.099 (5.1)	1.236 (39.9)
Very bad	765 (1.3)	339 (44.3)
Days with depression		
None	46.712 (77.6)	6.567 (14.1)
Less than half days	8.564 (14.2)	2.321 (27.1)
More than half days	2.472 (4.1)	805 (32.6)
Almost everyday	2.454 (4.1)	885 (36.1)
Smoking		
Never smoke	41.215 (80.0)	6.073 (14.7)
Less then daily	2.454 (4.8)	547 (22.3)
Daily basis	7.804 (15.2)	2.105 (26.0)
Sociodemographic conditions		
Low	18.864 (31.2)	3.419 (18.1)
Intermediate	22.297 (36.9)	4.009 (17.0)
High	19.300 (31.9)	3.419 (18.6)
Gini’s coefficient		
≤0.56	25.758 (42.8)	4.580 (17.3)
0.57 to 0.61	24.894 (41.3)	4.385 (17.6)
≥0.62	9.550 15.9)	1.613 (16.9)
Human Development Index (HDI)		
≥0.735	19.874 (44.0)	3.539 (82.9)
0.675 to 0.734	4.870 (10.8)	3.492 (17.6)
≤0.674	20.458 (45.3)	3.547 (17.3)
Family Health Support Team (NASF)		
≤0.946	21.966 (36.5)	3.566 (16.2)
0.947 to 1.876	18.974 (31.5)	3.414 (17.0)
≥1.876	19.262 (32.0)	3.598 (18.7)

**Table 2 ijerph-22-00879-t002:** Multilevel Poisson regression for chronic back problems according to individual and contextual variables.

	Model Null	Model 1 ^1^ (n = 60,202)		Model 2 ^2^ (n = 60,202)		Final Model (n = 60,202) ^3^	
Variables	(n = 60,202)	RP (95%IC)	*p* Value	RP (95%IC)	*p* Value	RP (95%IC)	*p* Value
**1° Level (individual)**							
Sex							
Male		1		1		1	
Female		1.23 (1.15; 1.30)	<0.001	1.23 (1.15; 1.30)	<0.001	1.23 (1.15; 1.30)	<0.001
Age (years)							
≤33		1		1		1	
34 to 48		1.45 (1.35; 1.56)	<0.001	1.45 (1.35; 1.56)	<0.001	1.45 (1.35; 1.46)	<0.001
≥49		1.75 (1.61; 1.90)	<0.001	1.75 (1.61; 1.90)	<0.001	1.75 (1.61; 1.90)	<0.001
Heavy work activity							
Não		1		1		1	
Sim		1.37 (1.28; 1.46)	<0.001	1.37 (1.28; 1.46)	<0.001	1.37 (1.28; 1.46)	<0.001
Health state self perception							
Very good		1		1		1	
Good		1.51 (1.34; 1.69)	<0.001	1.51 (1.35; 1.70)	<0.001	1.51 (1.35; 1.70)	<0.001
Regular		2.54 (2.25; 2.87)	<0.001	2.55 (2.26; 2.88)	<0.001	2.55 (2.26; 2.88)	<0.001
Bad		3.45 (2.94; 4.04)	<0.001	3.46 (2.95; 4.05)	<0.001	3.46 (2.96; 4.05)	<0.001
Very bad		3.92 (3.03; 5.06)	<0.001	3.92 (3.03; 5.07)	<0.001	3.92 (3.03; 5.07)	<0.001
Days with depression							
None		1		1		1	
Less than half days		1.52 (1.41; 1.63)	<0.001	1.52 (1.41; 1.63)	<0.001	1.51 (1.41; 1.63)	<0.001
More than half days		1.70 (1.51; 1.92)	<0.001	1.70 (1.51; 1.92)	<0.001	1.70 (1.51; 1.92)	<0.001
Almost everyday		1.71 (1.50; 1.94)	<0.001	1.70 (1.50; 1.94)	<0.001	1.70 (1.50; 1.94)	<0.001
Smoking							
Never smoke		1		1		1	
Less than daily		1.24 (1.10; 1.40)	<0.001	1.24 (1.10; 1.40)	<0.001	1.24 (1.10; 1.40)	<0.001
Daily basis		1.37 (1.28; 1.48)	<0.001	1.37 (1.27; 1.48)	<0.001	1.37 (1.27; 1.48)	<0.001
**2° Level**							
Sociodemografic conditions							
Low				1		1	
Intermediate				0.98 (0.69; 1.40)	0.916	1.00 (0.78; 1.30)	0.980
High				1.09 (0.73; 1.63)	0.672	1.10 (0.77; 1.59)	0.577
Gini’s coefficient							
≥0.56				1		1	
0.57 to 0.61				1.00 (0.84; 1.21)	0.929	1.00 (0.84; 1.20)	0.959
≥0.62				0.96 (0.77; 1.19)	0.701	0.95 (0.77; 1.18)	0.664
Human Development Index (HDI)							
≥0.735				1		-	-
0.675 to 0.734				0.91 (0.63; 1.30)	0.588	-	-
≤0.674				0.79 (0.53; 1.18)	0.249	-	-
Family Health Support Team							
≤0.946				1		1	
0.947 to 1.876				1.16 (0.99; 1.38)	0.072	1.17 (1.00; 1.38)	0.048
≥ 1.876				1.28 (1.06; 1.54)	0.011	1.28 (1.07; 1.54)	0.007
**Aleatory Effects**							
Variance (95 %CI)	0.029 (0.016; 0.052)	0.030 (0.016; 0.058)		0.020 (0.010; 0.041)		0.020 (0.010; 0.041)	
Change in Variation (%)		59.8		37.4		0.0	
LR test (x^2^, *p* value)	218.42 (<0.001)	87.85 (<0.001)		54.98 (<0.001)		55.09 (<0.001)	

Note: ^1^ Model 1: multilevel multiple analysis involving variables at the individual level only. ^2^ Model 2: multilevel multiple analysis involving variables at the individual and contextual levels. ^3^ Model 3: final analysis of the model with the variables that presented significant ‘*p*’ values and fit the theoretical model.

## Data Availability

The data that support the findings of this study of the Health National Survey (PNS 2013) can be found in the National Health Research Report 2013, published by the Brazilian Institute of Geography and Statistics (IBGE), available at https://www.ibge.gov.br/estatisticas/sociais/justica-e-seguranca/29540-2013-pesquisa-nacional-de-saude.html?=&t=microdados, 13 January 2019.

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
