# Peer review of "Association of Individual and Contextual Factors with Chronic Spine Problems: An Analysis from the National Health Survey"

_ijerph, 2025, doi:10.3390/ijerph22060879_

Round 1

Reviewer 1 Report

Comments and Suggestions for Authors

Dear Author/s

I have highlighted several key points that I believe are important for presenting the study more solidly and clearly. I think implementing these revisions would significantly improve the quality of your study.

Introduction
The gap in knowledge that the study will fill should be more clearly emphasized. A clear and understandable hypothesis or assumption should be presented for your study. The terms “contextual factors” and “individual factors” are frequently used, but it is not clear which variables fall under these categories. It is important to clarify these concepts. Although literature is used in this section, many claims are too general and need to be supported by more current or international sources.
There are some spelling and language issues in the sentences that should be reviewed and corrected.
For example:
The expression “...with two levels of aggregation bank linkages” is unclear.
If applicable, the authors could use a clearer and more fluid purpose statement like the following:
This study aims to investigate how both individual-level (such as age, sex, occupation) and contextual-level (such as state-level health service coverage) factors are associated with the prevalence of chronic spine problems among Brazilian adults, using data from the 2013 National Health Survey.

Materials and Methods
There are many language and spelling mistakes, and the paragraph structures are fragmented. For example:
• “...variables chosen were sex and age group. The second level included heavy work activity...” Are there only two variables at the first level? This is confusing.
• “...analysis considered the complexity of the sampling.” This should be described with more technical terms.
• “The null model was initially carried out, in which variables with characteristics at the individual level were included.” No variables are included in the null model, so this statement is incorrect.
Especially, the modeling process is explained twice in the same way (lines 104-107 and 150-154).
How were classifications like “Sociodemographic conditions: Low, Intermediate and High” made? Which coefficients were used to form them? This should be detailed.
Inclusion and exclusion criteria should be added.

Discussion
This section interprets the study’s findings in a logical framework and relates them to the literature. However, the presentation is disorganized. Presenting the results in a regular flow is important for the clarity of the study.
The study limitations should be written more systematically. Additionally, more specific recommendations for future research should be provided. Although the need for longitudinal studies is mentioned, specific factors that should be tracked (such as depression levels, workload) should be specified.

Conclusion
This section should summarize the most prominent individual factors, and the results related to the social context should be expressed more carefully. Additionally, including a sentence or sentences with policy recommendations for health would be beneficial.

Author Response

Althea Yang

Mai 25, 2024.

Dear Althea Yang,

            Thank you for your email with the reviewers’ comments. We have reviewed the comments and edited the manuscript accordingly. Please, find attached our point-by-point response to the reviewers. All authors have read this protocol and agree with the International Journal of Environmental Research and Public Health policy. We hope the revised manuscript is now suitable for publication. 

Sincerely. Sanderson José Costa de Assis.

Editor and reviewer comments

Reviewer #1:

- 1) Introduction: The gap in knowledge that the study will fill should be more clearly emphasized. A clear and understandable hypothesis or assumption should be presented for your study. The terms “contextual factors” and “individual factors” are frequently used, but it is not clear which variables fall under these categories. It is important to clarify these concepts. Although literature is used in this section, many claims are too general and need to be supported by more current or international sources.

Response: Thank you for your comment. The suggestions were accepted. The foundation was increased, based on current literature on the interaction between individual and contextual factors.

- 2) Introduction. There are some spelling and language issues in the sentences that should be reviewed and corrected.

For example: The expression “...with two levels of aggregation bank linkages” is unclear.

Response: Thanks for your comment. The suggestions were accepted and replacements were made in the manuscript. This sentence was rewritten: "with two levels of aggregation across linked public databases", and the entire text was revised.

- 3) Introduction. If applicable, the authors could use a clearer and more fluid purpose statement like the following:

This study aims to investigate how both individual-level (such as age, sex, occupation) and contextual-level (such as state-level health service coverage) factors are associated with the prevalence of chronic spine problems among Brazilian adults, using data from the 2013 National Health Survey.

Response: Thanks for your comment. The suggestions were accepted and replacements were made in the manuscript. The objective was rewritten: "This study aims to investigate how both individual-level (such as age, sex, occupation) and contextual-level (such as state-level health service coverage) factors are associated with the prevalence of chronic spine problems among Brazilian adults, using data from the 2013 National Health Survey."

- 4) Materials and Methods. “...variables chosen were sex and age group. The second level included heavy work activity...” Are there only two variables at the first level? This is confusing.

Response: Thanks for your comment. The suggestions were accepted and replacements were made in the manuscript. the variables at each level were already described in topic 2.2. Context and research participants, but the text has been rewritten to make the information clearer:

"At the first level of aggregation, only two variables were included: sex and age group. At the second level, behavioral variables were considered: heavy work activity, perceived health status, number of days feeling depressed, and smoking addiction"

- 5) Materials and Methods. “...analysis considered the complexity of the sampling.” This should be described with more technical terms.

Response: Thanks for your comment. The suggestions were accepted and replacements were made in the manuscript. The text has been rewritten:

“All analyzes were conducted considering the complex sampling design of the 2013 PNS, using the Complex Sample module of SPSS software and the Survey Data Analysis command (svy prefix) in Stata 13.0, in order to correct for the effect that the clustering of primary sampling units has on the estimates, known as the design effect (DEFF).”

- 6) Materials and Methods. “The null model was initially carried out, in which variables with characteristics at the individual level were included.” No variables are included in the null model, so this statement is incorrect.

Especially, the modeling process is explained twice in the same way (lines 104-107 and 150-154).

Response: Thanks for your comment. The suggestions were accepted and corrections were made in the manuscript. The null model was carried out to verify the feasibility of multilevel modeling and did not include individual variables. Therefore, to avoid duplication in the text and to make the correction in the first citation of the null model, this part of the text was excluded in topic 2.4. Quantitative variables.

- 7) Materials and Methods. How were classifications like “Sociodemographic conditions: Low, Intermediate and High” made? Which coefficients were used to form them? This should be detailed.

Response: Thanks for your comment. The Sociodemographic condition variable was obtained through a factor analysis, in which total fertility rate, infant mortality, percentage of people vulnerable to poverty, per capita income, and formal employment status of individuals over 17 years old were used to transform them into a single variable that was called Sociodemographic condition, then its categorization was carried out through tertiles. This was found in the third paragraph of topic 2.5. Statistical Methods, however, it was rewritten for better understanding:

"Data on sociodemographic factors were derived through exploratory factor analysis, using the following variables: total fertility rate, infant mortality, percentage of people vulnerable to poverty, per capita income, and formal employment status of individuals over 17 years old. These variables were selected based on their theoretical relevance to ad-equately represent sociodemographic conditions, as well as their statistical association with both the outcome. Following the scree test, the Kaiser–Guttman criterion, and Horn's parallel analysis, only one factor was extracted. This factor was subsequently categorized into three groups based on tertiles to facilitate interpretation and analysis"

- 8) Materials and Methods: Inclusion and exclusion criteria should be added.

Response: Thanks for your comment. The suggestions were accepted and the following text was added in the Material and methods section:

“The population to be surveyed corresponds to residents of private households in Brazil, except those located in special census sectors (barracks, military bases, lodgings, camps, vessels, penitentiaries, penal colonies, prisons, jails, asylums, orphanages, convents and hospitals).”

- 9) Discussion. This section interprets the study’s findings in a logical framework and relates them to the literature. However, the presentation is disorganized. Presenting the results in a regular flow is important for the clarity of the study.

Response: Thanks for your comment. Your suggestion was accepted. For better understanding and regular sequence of results, a change was made in the order of presentation of the discussion.

- 10) Discussion. The study limitations should be written more systematically. Additionally, more specific recommendations for future research should be provided. Although the need for longitudinal studies is mentioned, specific factors that should be tracked (such as depression levels, workload) should be specified.

Response: Thanks for your comment. Your suggestion was accepted. The limitations of the study were better presented at the end of the discussion with greater detail of the methodological biases of the study.

- 11) Conclusion. This section should summarize the most prominent individual factors, and the results related to the social context should be expressed more carefully. Additionally, including a sentence or sentences with policy recommendations for health would be beneficial.

Response: Thanks for your comment. Your suggestion was accepted. The conclusion was rewritten, emphasizing the prominent individual factors and the contextual factors observed. An idea was added to create and strengthen a public policy that increases access to primary health care services, to reduce inequities with back problems.

Thank you for your comment. The manuscript has been revised accordingly.

Sincerely,

Sanderson José Costa de Assis. PT, Ph.D. Federal University of Rio Grande do Norte. Corresponding author. Natal, Rio Grande do Norte, Brazil.

Mobile: +5584996219425

e-mail: sandersonassis.fisio@gmail.com

Reviewer 2 Report

Comments and Suggestions for Authors

Abstract

  • The abstract should emphasize more clearly that the study is cross-sectional.

Introduction

  • Specify “pain in different regions of the body”.
  • Justify in more detail the research problem and the gaps based on existing literature.
  • More recent (post-2018) references are missing to update the epidemiological context.
  • To deepen the theoretical framework on social determinants of health to better contextualize the findings.

Results

  • The text tends to repeat what is already in the tables without adding interpretation.
  • The tables are extensive and could be simplified or integrated with information in the text.

Discussion and Conclusion

  • Explain and justify why lumbar lordosis can lead to further spinal problems. changes in structure or injury not necessarily accompanied by pain?
  • The possible interaction between individual and contextual factors is not explored in depth.
  • They should avoid statements that suggest causality.
  • Add some more limitations and future lines

Author Response

Mai 25, 2024.

Dear Althea Yang,

            Thank you for your email with the reviewers’ comments. We have reviewed the comments and edited the manuscript accordingly. Please, find attached our point-by-point response to the reviewers. All authors have read this protocol and agree with the International Journal of Environmental Research and Public Health policy. We hope the revised manuscript is now suitable for publication. 

Sincerely. Sanderson José Costa de Assis.

Editor and reviewer comments

Reviewer #2:

- 1) Abstract: The abstract should emphasize more clearly that the study is cross-sectional.

Response: Thanks for your comment. The suggestions were accepted and replacements were made in the manuscript. The study design was already indicated, but it was made clearer.

- 2) Introduction. Specify “pain in different regions of the body”..

Response: Thanks for your comment. The suggestions were accepted and correções were made in the manuscript.

- 3) Introduction. Justify in more detail the research problem and the gaps based on existing literature.

Response: Thanks for your comment. The suggestions were accepted. The introduction was rewritten and the existing gaps and theoretical basis were better detailed.

- 4) Introduction. More recent (post-2018) references are missing to update the epidemiological context.

Response: Thanks for your comment. The suggestions were accepted. The references were updated in the text and in the references.

- 5) Introduction. To deepen the theoretical framework on social determinants of health to better contextualize the findings.

Response: Thanks for your comment. The suggestions were accepted. The foundation was expanded, based on the current literature on the interaction between individual and contextual factors.

- 6) Results. The text tends to repeat what is already in the tables without adding interpretation.

Response: Response: Thanks for your comment. The text was rewritten in order to make it more concise and meet the recommendations. As for the interpretations, we believe that they were well considered in the discussion topic.

- 7) Results. The tables are extensive and could be simplified or integrated with information in the text.

Response: Thanks for your comment. The text was rewritten with the aim of integrating the information more directly into the tables. As for the size, because there are many variables and a complex analysis, it is not possible to reduce it without changing the variables and objectives of the study.

- 8) Discussion and Conclusion: Explain and justify why lumbar lordosis can lead to further spinal problems. changes in structure or injury not necessarily accompanied by pain?

Response: Thanks for your comment. This sentence was unclearly expressed and was rewritten as follows:

“in addition to hormonal factors related to pregnancy, such as relaxin, estrogen and pro-gesterone, increase the propensity for biomechanical changes, predisposing individuals to back problems. [3, 19, 20]”

- 9) Discussion and Conclusion. The possible interaction between individual and contextual factors is not explored in depth.

Response: Thanks for your comment. The discussion was rewritten, deepening the discussion on the topic.

- 10) Discussion and Conclusion. They should avoid statements that suggest causality.

Response: Thanks for your comment. The discussion was rewritten, removing texts that induced causality.

- 11) Discussion and Conclusion. Add some more limitations and future lines.

Response: Thanks for your comment. Your suggestion was accepted, we added more limitations and future lines of work at the end of the discussion.

Thank you for your comment. The manuscript has been revised accordingly.

Sincerely,

Sanderson José Costa de Assis. PT, Ph.D. Federal University of Rio Grande do Norte. Corresponding author. Natal, Rio Grande do Norte, Brazil.

Mobile: +5584996219425

e-mail: sandersonassis.fisio@gmail.com

Round 2

Reviewer 2 Report

Comments and Suggestions for Authors

most of the reviewer's suggestions have been met